# Fairness-Aware EHR Analysis via Structured Missing Pattern Modeling and Adversarial Low-Rank Adaptation

## Abstract

Deep learning has been widely applied to electronic health record (EHR) analysis, offering strong predictive capabilities for clinical outcome prediction. However, due to inherent disparities across demographic groups (e.g., race, gender), fairness concerns have become increasingly critical. Due to severe instability in jointly optimizing fairness and performance objectives, existing fairness-aware approaches often struggle to balance predictive accuracy and fairness. Moreover, the structured missingness in real-world EHR data further worsens the performance and fairness of predictive models, yet they are frequently overlooked in fairness-aware modeling. In this paper, we propose `FEMALA`, a novel two-stage EHR analysis framework by explicitly modeling the structured missing patterns within EHR and fairness-aware model adaptation. Particularly, we design a dual-encoder architecture to integrate EHR temporal dynamics and structured missing patterns, thus enhancing performance while improving fairness by handling missingness well. Further, we perform adversarial fine-tuning to decorrelate task and sensitive representations via low-rank adaptation, enabling a better trade-off between fairness and accuracy. Experiments on MIMIC-III/IV datasets demonstrate that our framework achieves state-of-the-art performance in both accuracy and fairness, validating the effectiveness of structured missing pattern modeling and fairness adaptation fine-tuning. The code is anonymously available at `https://anonymous.4open.science/r/FEMALA/README.md`.

## 1 Introduction

The widespread adoption of Electronic Health Records (EHRs) has enabled the collection of rich longitudinal patient data (Johnson et al., 2016; 2023; Pollard et al., 2018), fueling the development of data-driven models for clinical decision-making. These models show great promise in forecasting high-stakes outcomes like in-hospital mortality and readmission (Acosta et al., 2022; Hayat et al., 2022; Wang et al., 2024b), paving the way for more timely and personalized interventions.

However, the deployment of these models has raised significant concerns about algorithmic fairness, as they often exhibit biased performance across demographic groups (Huang et al., 2023; Tarek et al., 2024). This unfairness typically originates from historical biases in healthcare practices embedded within EHR data. For instance, underprivileged groups may receive less frequent care, leading to sparser, noisier data and consequently, worse prediction outcomes (Zhang et al., 2024). While many fairness-aware methods have been proposed—ranging from fairness regularization (Lin et al., 2023) to adversarial debiasing (Wang et al., 2024a)—they often struggle to balance predictive accuracy and fairness, and crucially, tend to overlook another fundamental challenge of EHR data: *missingness*.

Missing values are pervasive in EHRs, but are rarely random. Instead, they often exhibit *structured missingness*—systematic patterns that reflect variations in clinical workflows, patient conditions, or care disparities (Li et al., 2021; Getzen et al., 2023). For example, patients from certain demographic groups may be less likely to receive specific diagnostic tests. This presents a critical, yet underexplored, challenge at the intersection of fairness and missing data. Most prior work treats these as two separate problems: imputation methods (Du et al., 2023) aim to fill in missing values but can inadvertently mask or even amplify the very biases encoded in the missingness patterns,

while fairness methods typically assume complete or pre-imputed data, ignoring that the pattern of missingness itself is a potent source of bias (Zhang and Long, 2021; Caton et al., 2022). This oversight leaves a critical gap: how can we build fair predictive models when the very structure of data absence is intertwined with demographic disparities?

To address this challenge, we argue that structured missingness should not be treated as noise to be eliminated, but as a rich, informative signal to be explicitly modeled. We propose FEMALA (**F**airness-Aware **E**HR Analysis via **M**issing Pattern Modeling and **A**dversarial **L**ow-Rank **A**daptation), a novel two-stage framework that directly confronts the interplay between structured missingness and fairness. In the first stage, we introduce a dual-encoder architecture that simultaneously learns from both the observed clinical events and the patterns of missingness. By adaptively fusing these two heterogeneous information streams, FEMALA develops a more robust and nuanced patient representation that mitigates biases arising from data collection disparities. In the second stage, we employ a stable and parameter-efficient adversarial fine-tuning strategy using low-rank adaptation (LoRA). This "learn first, correct later" approach allows us to first build a strong predictive model grounded in a comprehensive understanding of the data, and then precisely remove residual correlations with sensitive attributes, achieving a superior trade-off between accuracy and fairness.

Our contributions are summarized as follows: **(1)** We identify and address the critical, underexplored problem of how structured missingness in EHRs exacerbates algorithmic bias, proposing a novel two-stage framework, FEMALA, to tackle this challenge; **(2)** We design a dual-encoder architecture that explicitly models both temporal EHR signals and structured missingness patterns, integrating them via a novel fusion module to learn fairer and more robust representations; **(3)** We introduce a stable adversarial low-rank adaptation strategy for fairness fine-tuning, which effectively improves fairness with minimal impact on predictive accuracy; **(4)** FEMALA achieves state-of-the-art performance on MIMIC-III and MIMIC-IV datasets, demonstrating significant gains in both accuracy (e.g., a 2.3% AUROC gain on MIMIC-III) and fairness (e.g., a 2.9% EO reduction) over the strongest baselines.

## 2 RELATED WORK

**Fairness-Aware Predictive Models in Healthcare.** Algorithmic fairness is crucial in healthcare, where biased predictions can worsen disparities and jeopardize patient safety Creager et al. (2019); Oh et al. (2022); Pessach and Shmueli (2022); Park et al. (2022; 2021); Li et al. (2022); Ktena et al. (2024); Meng et al. (2022). A broad spectrum of fairness-aware strategies has been proposed to mitigate demographic bias, including loss-based regularization Lin et al. (2023); Sivarajkumar et al. (2023), adversarial debiasing that suppresses sensitive information in latent representations Yang et al. (2023); Xu et al. (2024); Luo et al. (2024); Poulain et al. (2023); Wang et al. (2024a), and counterfactual approaches leveraging disentangled representations Liu et al. (2023; 2022) or contrastive learning Wang et al. (2024c). Despite recent advances, most methods assume fully observed or well-imputed data—an assumption that rarely holds in real-world EHRs with pervasive, structured missingness. Moreover, single-stage optimization of accuracy and fairness often yields unstable convergence and brittle trade-offs, as early fairness constraints may suppress informative but biased signals.

**Handling Missing Data in Electronic Health Records.** Missingness in EHRs is not only pervasive but also highly structured—often following non-random patterns that correlate with clinical relevance and sensitive attributes Mitra et al. (2023). For example, patients from underserved groups may receive fewer diagnostic tests, resulting in systematic data gaps that encode structural bias. Standard approaches rely on imputation Wu et al. (2022); Fortuin et al. (2020); Cao et al. (2018); Yoon et al. (2018), which may obscure meaningful patterns or exacerbate disparities Jeanselme et al. (2022); Goel et al. (2021). Recent work has begun addressing fairness under missingness, introducing imputation-free models Jeong et al. (2022), graph-based debiasing Guo et al. (2023), and missingness-aware classifiers Feng et al. (2023). However, these methods are typically designed for low-dimensional or tabular data, and do not generalize well to high-dimensional, temporally structured EHRs, where missingness itself reflects complex, bias-related structure that remains underexplored.

**Low-Rank Adaptation for Fine-Tuning.** Parameter-efficient fine-tuning (PEFT) methods such as LoRA Hu et al. (2022) adapt large pretrained models by updating only a small set of low-rank parameters. While initially proposed for efficiency, LoRA has been extended to fairness-critical

tasks Das et al. (2024); Liu et al.; Ding et al. (2024). For example, FairLoRA Sukumaran et al. (2024) adds fairness-specific regularization, FairTune Dutt et al. uses bi-level optimization to adjust adapter masks, and other approaches remove bias without relying on sensitive attributes Kamalaruban et al. (2025). However, LoRA remains largely unexplored in clinical prediction, where balancing fairness and accuracy is particularly challenging due to the high dimensionality and structured missingness of EHR data.

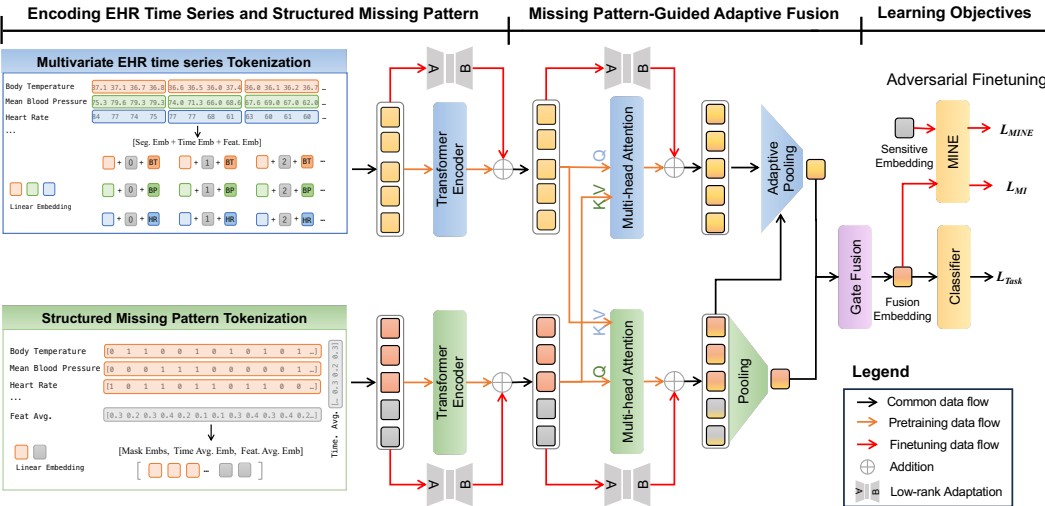

Figure 1: Overview of our FEMALA framework. Stage I encodes EHR sequences and structured missingness via a Segment-Aware Temporal Encoder and a Structured Missingness Encoder, which are fused through a Missingness-Guided Adaptive Fusion module. Stage II applies adversarial fine-tuning on low-rank adapters to enforce fairness by minimizing mutual information between task and sensitive representations.

## 3 METHOD

### 3.1 PRELIMINARIES AND FRAMEWORK OVERVIEW

**Problem Formulation.** Consider a cohort of $N$ patients, where each admission is represented as a multivariate time series $\mathbf{X} = [x_{t,d}]_{t=1,d=1}^{T,D} \in \mathbb{R}^{T \times D}$, with $T$ denoting the number of time steps and $D$ the number of clinical variables. A corresponding binary mask $\mathbf{M} \in \{0,1\}^{T \times D}$ indicates missing entries, where $m_{t,d} = 0$ if $x_{t,d}$ is unobserved. Each patient is also associated with $k$ sensitive attributes $\mathbf{A} = [a_1, \ldots, a_k]$, where $a_k \in \{0,1\}^{d_k}$ denotes a one-hot encoding of attribute categories (e.g., *race*, *gender*). Our goal is to learn a predictive model $f_\Theta : (\mathbf{X}, \mathbf{M}) \mapsto \hat{y} \in \mathcal{Y}$ that estimates a clinical outcome $y$ (e.g., in-hospital mortality), while minimizing performance disparities across subgroups defined by $\mathbf{A}$.

**Low-Rank Adaptation.** Low-Rank Adaptation (LoRA) (Hu et al., 2022) enables parameter-efficient fine-tuning by injecting low-rank updates into frozen pretrained weights. For a projection matrix $W \in \mathbb{R}^{d_{\text{out}} \times d_{\text{in}}}$, LoRA introduces a trainable update $\Delta W = BA$, where

$$W \leftarrow W + \Delta W, \quad A \in \mathbb{R}^{r \times d_{\text{in}}}, \quad B \in \mathbb{R}^{d_{\text{out}} \times r}, \quad r \ll \min(d_{\text{out}}, d_{\text{in}}).$$

This reduces trainable parameters from $d_{\text{out}}d_{\text{in}}$ to $r(d_{\text{out}} + d_{\text{in}})$, and incurs no inference overhead, as $\Delta W$ can be merged post-training.

**Overview of Our Framework.** Figure 1 illustrates the overall framework. FEMALA follows a two-stage strategy combining pretraining and fine-tuning to achieve fairness-aware EHR prediction. In the first stage, we enhance representation learning by jointly modeling temporal patterns and structured missingness. A segment-aware temporal encoder and a structured missingness encoder extract complementary features, which are fused via a missing pattern-guided adaptive fusion module

to improve both accuracy and fairness while properly handling the missing data. In the second stage, we obtain the sensitive attribute embedding through an autoencoder-based architecture, and further estimate the mutual information (i.e., dependence) between these embeddings and task representations. We apply adversarial fine-tuning via low-rank adaptation to reduce this dependence, achieving stable fairness improvements with minimal impact on predictive performance.

## 3.2 Enhanced EHR Representation with Structured Missing Pattern Learning

**Segment-Aware Temporal Encoder.** To capture the temporal dynamics of multivariate time-series EHR data $\mathbf{X} \in \mathbb{R}^{T \times D}$, we divide it into non-overlapping segments of fixed length $L$ (e.g., 4 hours), which demonstrated effective in time-seiries process (Zhang and Yan, 2023). The segmented input is reshaped into $\{\mathbf{x}_{s,d}\}_{s=1,d=1}^{S,D}$, where $S = T/L$ and each segment $\mathbf{x}_{s,d} \in \mathbb{R}^L$ corresponds to variable $d$ in segment $s$. Each segment is projected into a $d_h$-dimensional embedding space via a shared linear projection

$$\mathbf{h}_{s,d} = \mathbf{W}_{seg}\mathbf{x}_{s,d}, \quad \mathbf{W}_{seg} \in \mathbb{R}^{d_h \times L}. \tag{1}$$

To encode both temporal and feature-wise semantics, we add learnable positional embeddings along two axes

$$\mathbf{h}_{s,d} = \mathbf{h}_{s,d} + \mathbf{p}_{\text{time}}(s) + \mathbf{p}_{\text{feat}}(d), \quad \mathbf{p}_{\text{time}}, \mathbf{p}_{\text{feat}} \in \mathbb{R}^{d_h}. \tag{2}$$

Here, $\mathbf{p}_{\text{time}}(s)$ encodes the temporal position of segment $s$, and $\mathbf{p}_{\text{feat}}(d)$ encodes the identity of clinical variable $d$. The resulting tokens are reshaped into a sequence $\mathbf{H}_{\text{seg}} \in \mathbb{R}^{B \times SD \times d_h}$, where $B$ is the batch size. A transformer encoder Vaswani et al. (2017) is then applied to capture contextual interactions across time and variables

$$\mathbf{E}_{\text{seg}} = \text{Transformer}(\mathbf{H}_{\text{seg}}), \quad \mathbf{E}_{\text{seg}} \in \mathbb{R}^{B \times SD \times d_h}. \tag{3}$$

**Structured Missing Pattern Encoder.** To model the structured missingness patterns, we enrich the time point-level missingness mask $\mathbf{M}$ with global missing pattern information. Specifically, we compute two summary vectors: the time-wise average missingness per feature and the feature-wise average missingness per time step as

$$\mathbf{m}_{\text{time}} = \left[\frac{1}{T}\sum_{t=1}^{T} m_{t,d}\right]_{d=1}^{D} \in \mathbb{R}^D, \quad \mathbf{m}_{\text{feat}} = \left[\frac{1}{D}\sum_{d=1}^{D} m_{t,d}\right]_{t=1}^{T} \in \mathbb{R}^T. \tag{4}$$

After that, we concatenate these global missingness with the original mask to form $\mathbf{M}' = [\mathbf{M}, \mathbf{m}_{\text{time}}, \mathbf{m}_{\text{feat}}]$. An $1 \times 1$ convolution is then, respectively, applied to project the tokens into a shared embedding space,

$$\mathbf{H}_{\text{mask}} \in \mathbb{R}^{B \times N_{\text{mask}} \times d_h}, \quad N_{\text{mask}} = T + 2,$$

where $B$ is the batch size and $d_h$ is the embedding dimension. Finally, a Transformer encoder is used to model contextual dependencies among the mask tokens, yielding

$$\mathbf{E}_{\text{mask}} = \text{Transformer}(\mathbf{H}_{\text{mask}}) \in \mathbb{R}^{B \times N_{\text{mask}} \times d_h}.$$

**Missing Pattern-Guided Adaptive Fusion.** After obtaining the segment-aware EHR embeddings $\mathbf{E}_{\text{seg}}$ and the structured missingness embeddings $\mathbf{E}_{\text{mask}}$, we apply a dual cross-attention mechanism to enable mutual enrichment between the two representations

$$\tilde{\mathbf{E}}_{\text{seg}} = \text{MHA}(\mathbf{E}_{\text{seg}}, \mathbf{E}_{\text{mask}}), \quad \tilde{\mathbf{E}}_{\text{mask}} = \text{MHA}(\mathbf{E}_{\text{mask}}, \mathbf{E}_{\text{seg}}), \tag{5}$$

where MHA denotes multi-head attention. To obtain compact representations, we further apply mean pooling to $\tilde{\mathbf{E}}_{\text{mask}}$, yielding a summary vector $\mathbf{z}_{\text{mask}} \in \mathbb{R}^{d_h}$. For the segment embeddings $\tilde{\mathbf{E}}_{\text{seg}}$, we employ adaptive attention pooling guided by the learned missing pattern. Specifically, attention weights $\alpha \in [0,1]^{B \times SD}$ are computed using the mask embedding

$$\alpha_i = \text{softmax}\left(\mathbf{W_a}\tilde{\mathbf{E}}_{\text{mask},i}\right), \quad \mathbf{W_a} \in \mathbb{R}^{SD \times N_{\text{mask}}}. \tag{6}$$

The weighted sum then produces the aggregated segment-level embedding

$$\mathbf{z}_{\text{seg}} = \sum_i \alpha_i \cdot \tilde{\mathbf{E}}_{\text{seg},i}. \tag{7}$$

Finally, we combine $\mathbf{z}_{\text{seg}}$ and $\mathbf{z}_{\text{mask}}$ using a learnable gating mechanism

$$\mathbf{z} = \mathbf{g} \odot \mathbf{z}_{\text{seg}} + (1 - \mathbf{g}) \odot \mathbf{z}_{\text{mask}}, \quad \mathbf{g} = f(\mathbf{z}_{\text{seg}} \oplus \mathbf{z}_{\text{mask}}), \tag{8}$$

where $\odot$ denotes element-wise multiplication, $\oplus$ denotes vector concatenation, and $f$ is an MLP-based gating function. The final fused representation $\mathbf{z}$ is passed to an MLP classifier to produce the final prediction $\hat{y}$.

### 3.3 ADVERSARIAL LOW-RANK FINE-TUNING FOR FAIRNESS-AWARE ANALYSIS

**Learning Sensitive Attribute Embeddings.** We map the sensitive attributes $\mathbf{A}$ into the same latent space as the task representation $\mathbf{z}$, so that we can estimate the mutual information between them. As illustrated in Figure 2, we adopt an autoencoder-based architecture to obtain sensitive attributes embedding $\mathbf{e}_s$. Each sensitive attribute is further encoded into a binary vector $\mathbf{a}_k \in \mathbb{R}^{d_k}$, where $d_k$ is the number of subgroups in attribute $k$. The concatenated vector $\mathbf{A} = [\mathbf{a}_1, \ldots, \mathbf{a}_k]$ serves as the input to an encoder $f_\psi$, implemented as a two-layer MLP

$$\mathbf{e}_s = f_\psi(\mathbf{A}) \in \mathbb{R}^{d_h}. \tag{9}$$

To ensure $\mathbf{e}_s$ retains full information about the sensitive attributes, we reconstruct the sensitive attribute via attribute-specific decoders $\mathcal{D}_k : \mathbb{R}^{d_h} \to \mathbb{R}^{d_k}$,

$$\hat{\mathbf{a}}_k = \text{softmax}(\mathcal{D}_k(\mathbf{e}_s)). \tag{10}$$

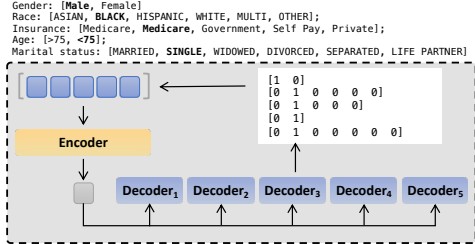

Figure 2: Sensitive attribute embedding learning architecture.

**Mutual Information Estimation.** To quantify the dependence between the sensitive attribute embedding $\mathbf{e}_s$ and the task representation $\mathbf{z}$, we adopt the Mutual Information Neural Estimation (MINE) framework (Belghazi et al., 2018). A neural critic network $T_\phi$ is trained to distinguish between joint samples from $p(\mathbf{z}, \mathbf{e}_s)$ and independent samples from the product of marginals $p(\mathbf{z})p(\mathbf{e}_s)$. The mutual information is estimated as

$$\widehat{\mathcal{I}}_\phi(\mathbf{z}; \mathbf{e}_s) = \mathbb{E}_{p(\mathbf{z}, \mathbf{e}_s)}[T_\phi(\mathbf{z}, \mathbf{e}_s)] - \log \mathbb{E}_{p(\tilde{\mathbf{z}})p(\mathbf{e}_s)}\left[e^{T_\phi(\tilde{\mathbf{z}}, \mathbf{e}_s)}\right], \tag{11}$$

where $\tilde{\mathbf{z}}$ denotes a shuffled or independently sampled task embedding. The first term approximates the expected critic score under the joint distribution, while the second approximates the score under the independent baseline. The critic $T_\phi$ is optimized to maximize this difference, thereby producing a tight lower bound on the true mutual information.

**Adversarial Fine-Tuning with Low-Rank Adaptation.** To achieve fairness, we perform adversarial fine-tuning over the low-rank adapters $\Delta\theta$, encouraging the final representation $\mathbf{z}'$ to be invariant to sensitive attributes. The training objective is formulated as a minimax game: The mutual information estimator $T_\phi$ is trained to *maximize* $\widehat{\mathcal{I}}_\phi(\mathbf{z}'; \mathbf{e}_s)$, improving the estimation quality. Simultaneously, the low-rank adapters $\Delta\theta$ are trained to *minimize* $\widehat{\mathcal{I}}_\phi(\mathbf{z}'; \mathbf{e}_s)$, while preserving task performance. This adversarial objective encourages the model to learn task-relevant representations decorrelated from sensitive attributes.

### 3.4 OBJECTIVES

In the pre-training stage, we optimize the representation framework via a binary cross-entropy loss for clinical outcome prediction

$$\mathcal{L}_{\text{task}} = -\frac{1}{N} \sum_{i=1}^N [y_i \log \hat{y}_i + (1 - y_i) \log(1 - \hat{y}_i)], \quad \hat{y}_i = f_\theta(\mathbf{X}_i, \mathbf{M}_i). \tag{12}$$

In the second stage, the overall adversarial fine-tuning objective for updating low-rank adapters $\Delta\theta$ can be represented as

$$\mathcal{L}_{\text{adv}} = -\frac{1}{N} \sum_{i=1}^N [y_i \log \hat{y}_i' + (1 - y_i) \log(1 - \hat{y}_i')] + \lambda_{\text{MI}} \widehat{\mathcal{I}}_\phi(\mathbf{z}; \mathbf{e}_s), \quad \hat{y}_i' = f_{\theta+\Delta\theta}(\mathbf{X}_i, \mathbf{M}_i), \tag{13}$$

where $\lambda_{\text{MI}}$ controls the mutual information regularization strength and is empirically set to 0.5.

Table 1: Summary statistics of the datasets used.

| Dataset | MIMIC-III | | | MIMIC-IV | | |
| Split | Training | Validation | Test | Training | Validation | Test |
| --- | --- | --- | --- | --- | --- | --- |
| Total | 12,672 | 1,833 | 3,638 | 15,112 | 2,188 | 4,473 |
| Missing rate | 72.81% | 72.71% | 72.73% | 71.16% | 71.17% | 71.23% |
| Positive (**IHM**) | 1,481 | 237 | 446 | 1,702 | 243 | 511 |
| Positive (**READM**) | 2,268 | 364 | 647 | 2,648 | 390 | 814 |

## 4 EXPERIMENT

### 4.1 EXPERIMENTAL SETUP

**Datasets.** We evaluate `FEMALA` on two large-scale, real-world EHR datasets: MIMIC-III (Johnson et al., 2016) and MIMIC-IV (Johnson et al., 2023). Both datasets contain de-identified health records of patients admitted to intensive care units (ICUs) or emergency departments at Beth Israel Deaconess Medical Center (BIDMC), comprising multivariate time-series data (e.g., vital signs, lab tests) and demographic information. Following established preprocessing protocols (Harutyunyan et al., 2019; Wang et al., 2024c), we extract 26 continuous-valued clinical variables from MIMIC-III and 25 from MIMIC-IV, sampled hourly during the first 48 hours of ICU admission. Table 1 summarizes dataset statistics. We obtain 18,143 ICU stays from MIMIC-III and 21,773 from MIMIC-IV, randomly splitting each into training, validation, and test sets using a 7:1:2 ratio. Additional preprocessing and variable details are provided in Appendix C.

**Tasks & Evaluation Metrics.** Following common practices in clinical prediction (Hayat et al., 2022; Wu et al., 2024; Wang et al., 2024c; Zhang et al., 2022), we evaluate `FEMALA` on two widely studied prediction tasks: (1) *In-Hospital Mortality (IHM) prediction*, which identifies whether a patient will die during hospitalization; and (2) *Readmission (READM) prediction*, which assesses whether a patient will be readmitted within 30 days of discharge. We report predictive performance metrics including AUROC, AUPR, and F1 score. To evaluate group fairness, we adopt Equalized Odds (EO) and Error Distribution Disparity Index (EDDI) following (Wang et al., 2024c). Fairness metric definitions are provided in Appendix C.2. All reported metrics include 95% confidence intervals computed via 1,000 bootstrap samples.

**Implementation Details.** We train all models using the Adam optimizer Kingma and Ba (2014) with a learning rate of 4e-4 and a batch size of 128. Early stopping is employed if the validation AUROC does not improve for 10 consecutive epochs. All experiments are conducted on a single NVIDIA RTX 4090 GPU. Further details are in Appendix D.1.

**Compared Methods.** We compare `FEMALA` with three categories of baselines: (1) *Backbone-only models without fairness constraints*: Transformer Vaswani et al. (2017), LSTM Graves and Graves (2012), RNN Elman (1990), CNN LeCun et al. (1998); (2) *General fairness-aware models*: FFVAE Creager et al. (2019), FarconVAE Oh et al. (2022); and (3) *Clinical fairness baselines*: FairEHR-CLP Wang et al. (2024c), FLMD Liu et al. (2023). See Appendix D.2 for details.

### 4.2 PERFORMANCE AND FAIRNESS ANALYSIS

**Overall Performance.** Table 2 summarizes the main results. Our analysis proceeds in two steps, mirroring our two-stage framework. **First, in the pre-training (PT) stage**, our model (PT) already establishes a new state-of-the-art in both predictive accuracy and fairness, outperforming all backbone models. By explicitly modeling structured missingness, our approach achieves a 2.2% AUROC gain on MIMIC-III IHM and, crucially, also reduces fairness disparities without any explicit fairness supervision. This result provides strong evidence for our core hypothesis: properly handling structured missingness is a critical first step towards achieving algorithmic fairness. **Second, in the fine-tuning (FT) stage**, it further enhances fairness, achieving the best fairness metrics (EO and EDDI) across nearly all settings. In contrast to baselines like FLMD, which trade accuracy for fairness, our adversarial low-rank tuning preserves high predictive performance while substantially reducing bias. This demonstrates the stability and effectiveness of our "learn first, correct later" strategy. Although

Table 2: Performance and fairness evaluation across two datasets: MIMIC-III and MIMIC-IV. All results(%) are reported with 95% confidence intervals. The best results are highlighted in **bold**, and the second-best results are underlined. Avg. Rank indicates the average ranking of each method across five evaluation metrics.

| Model | In-Hospital Mortality | | | | | | Readmission | | | | | |
|---|---|---|---|---|---|---|---|---|---|---|---|---|
| | AUROC (↑) | AUPR (↑) | F1 (↑) | EO (↓) | EDDI (↓) | Avg. Rank | AUROC (↑) | AUPR (↑) | F1 (↑) | EO (↓) | EDDI (↓) | Avg. Rank |
| **Dataset 1: MIMIC-III** | | | | | | | | | | | | |
| Transformer Vaswani et al. (2017) | 80.49(78.58, 82.48) | 38.54(33.95, 43.61) | 41.09(37.64, 44.56) | 10.09(8.77, 14.54) | 4.13(3.56, 5.91) | 4.6 | 70.26(67.85, 72.49) | 38.87(35.09, 43.15) | 40.43(37.51, 43.36) | 8.97(8.55, 13.51) | 6.05(4.67, 8.68) | 4.4 |
| LSTM (Graves and Graves, 2012) | 82.41(80.55, 84.38) | 42.34(37.66, 47.69) | 43.76(40.56, 47.19) | 8.77(8.53, 13.81) | 4.76(3.98, 6.46) | 3.6 | 72.39(70.17, 74.71) | 39.35(35.67, 43.51) | 41.58(38.84, 44.5) | 8.12(7.97, 12.95) | 6.15(4.62, 8.57) | 3 |
| RNN Elman (1990) | 81.39(79.31, 83.50) | 43.38(38.45, 48.06) | 43.26(39.75, 46.98) | 8.32(8.08, 13.56) | 4.06(3.47, 6.02) | 3 | 71.60(69.41, 73.81) | 38.93(35.15, 42.91) | 40.88(37.94, 43.57) | 9.62(8.77, 14.29) | 4.63(3.98, 7.57) | 3.8 |
| CNN (LeCun et al., 1998) | 82.28(80.39, 84.17) | 44.19(39.37, 49.07) | 44.03(40.35, 47.47) | 8.48(8.33, 13.19) | 4.58(3.69, 6.11) | 2.8 | 74.15(72.14, 76.30) | 42.55(38.50, 46.78) | 43.18(40.42, 46.00) | 9.46(8.32, 13.75) | 5.57(4.32, 8.29) | 2.6 |
| Ours(PT) | 84.59(82.71, 86.38) | 48.22(43.51, 53.05) | 47.88(44.41, 51.17) | 6.78(7.48, 12.67) | 3.28(2.97, 5.46) | 1 | 76.49(74.39, 78.61) | 47.00(42.80, 51.27) | 46.34(43.54, 49.22) | 8.85(8.72, 13.84) | 4.60(3.47, 7.29) | 1.2 |
| FairEHR-CLP (Wang et al., 2024c) | 79.70(77.63, 81.86) | 35.83(31.61, 40.29) | 40.35(36.79, 43.75) | 8.46(8.30, 13.63) | 4.32(3.46, 6.31) | 4.6 | 73.72(71.41, 75.98) | 38.96(35.05, 43.30) | 41.65(39.09, 43.97) | 8.59(8.28, 14.11) | 6.10(5.26, 9.32) | 3.8 |
| FLMD (Liu et al., 2023) | 81.77(79.74, 83.72) | 41.72(37.13, 46.91) | 43.51(39.87, 47.12) | 9.73(8.95, 14.54) | 4.32(3.80, 5.83) | 3.4 | 73.27(71.13, 75.48) | 41.06(37.26, 45.40) | 42.18(39.41, 44.84) | 7.87(7.28, 13.26) | 5.72(4.53, 8.17) | 2.8 |
| FFVAE (Creager et al., 2019) | 82.19(80.27, 84.06) | 41.08(36.23, 45.85) | 41.73(38.08, 45.26) | 6.86(6.32, 11.08) | 3.35(2.84, 5.42) | 3 | 71.84(69.48, 73.96) | 37.50(33.83, 41.71) | 40.57(37.92, 42.95) | 8.05(7.56, 12.98) | 4.53(3.47, 7.37) | 4.2 |
| FarconVAE (Oh et al., 2022) | 82.27(80.21, 84.10) | 40.80(35.96, 45.31) | 42.93(39.17, 46.56) | 7.38(5.10, 15.42) | 2.40(1.50, 4.05) | 2.8 | 73.70(71.52, 75.83) | 39.56(35.80, 43.81) | 42.05(38.91, 45.14) | 8.99(6.11, 16.04) | 3.82(2.48, 7.00) | 3.2 |
| Ours (FT) | 83.49(81.63, 85.42) | 46.66(41.93, 51.38) | 48.18(44.68, 51.39) | 5.37(5.04, 10.26) | 1.95(1.84, 4.10) | 1 | 74.60(72.34, 76.83) | 45.44(41.17, 49.59) | 44.55(41.57, 47.65) | 4.92(4.18, 9.21) | 2.80(2.24, 5.29) | 1 |
| **Dataset 2: MIMIC-IV** | | | | | | | | | | | | |
| Transformer Vaswani et al. (2017) | 82.35(80.61, 84.16) | 40.98(36.80, 45.71) | 42.94(39.56, 46.23) | 5.11(4.73, 9.03) | 3.02(2.54, 4.22) | 4 | 71.96(69.92, 73.95) | 39.87(36.55, 43.65) | 42.05(39.22, 45.00) | 6.20(5.43, 8.90) | 3.20(2.70, 5.01) | 3.8 |
| LSTM (Graves and Graves, 2012) | 82.93(81.08, 84.83) | 45.50(40.89, 49.95) | 43.50(40.08, 46.60) | 5.27(5.22, 8.55) | 3.20(2.71, 4.26) | 3.8 | 73.09(71.10, 75.00) | 43.62(39.94, 46.85) | 42.98(40.35, 45.39) | 7.52(6.18, 10.63) | 3.69(3.05, 5.66) | 3.4 |
| RNN Elman (1990) | 82.39(80.46, 84.08) | 43.65(39.37, 48.06) | 44.18(40.52, 47.79) | 5.69(4.88, 9.29) | 2.53(2.05, 4.32) | 3.6 | 72.66(70.75, 74.60) | 42.32(38.75, 45.87) | 43.20(40.88, 45.65) | 6.00(5.62, 9.33) | 3.78(2.94, 5.96) | 3.2 |
| CNN (LeCun et al., 1998) | 83.56(81.66, 85.27) | 46.65(42.02, 51.09) | 44.57(40.93, 48.03) | 5.18(5.30, 8.37) | 3.03(2.50, 4.02) | 2.6 | 73.74(71.89, 75.66) | 42.46(38.93, 45.94) | 44.07(41.29, 46.46) | 7.26(6.45, 10.41) | 3.81(2.91, 5.75) | 3.2 |
| Ours(PT) | 84.04(82.30, 85.65) | 47.12(42.53, 51.31) | 45.89(42.18, 49.07) | 3.40(3.75, 7.64) | 2.31(1.87, 3.45) | 1 | 73.74(72.88, 76.47) | 45.12(41.86, 48.68) | 44.94(42.24, 47.63) | 6.05(5.29, 9.30) | 3.33(2.65, 5.12) | 1.4 |
| FairEHR-CLP (Wang et al., 2024c) | 83.45(81.65, 85.14) | 42.07(37.63, 46.57) | 40.48(36.38, 44.17) | 5.84(5.20, 9.54) | 2.35(2.12, 3.53) | 4 | 74.36(72.39, 76.32) | 41.62(37.93, 45.08) | 42.60(39.73, 45.40) | 7.27(6.20, 9.90) | 3.11(2.46, 4.85) | 4 |
| FLMD (Liu et al., 2023) | 83.69(81.78, 85.50) | 44.80(40.16, 49.45) | 43.44(39.47, 47.33) | 3.63(3.46, 7.39) | 2.36(1.92, 3.37) | 2.6 | 74.24(72.23, 76.14) | 42.43(38.68, 46.25) | 42.87(39.83, 46.03) | 4.03(3.75, 7.01) | 2.69(2.25, 4.22) | 2.6 |
| FFVAE (Creager et al., 2019) | 82.46(80.54, 84.28) | 41.44(36.72, 46.22) | 42.74(38.59, 46.48) | 4.55(4.05, 8.02) | 2.15(1.70, 3.18) | 3.6 | 73.20(71.16, 75.14) | 41.99(38.45, 45.85) | 42.37(39.61, 45.26) | 4.47(3.96, 7.23) | 2.70(2.36, 4.45) | 4.2 |
| FarconVAE (Oh et al., 2022) | 83.14(81.30, 84.88) | 40.30(35.66, 45.10) | 42.54(38.74, 45.86) | 4.78(2.92, 7.76) | 1.74(1.24, 2.69) | 3.6 | 73.42(71.46, 75.37) | 42.47(38.99, 46.01) | 44.01(41.21, 47.08) | 4.87(3.15, 7.51) | 2.40(1.54, 3.47) | 2.6 |
| Ours (FT) | 83.91(82.16, 85.47) | 46.36(41.69, 50.44) | 45.81(42.35, 49.13) | 2.39(3.51, 7.19) | 1.77(1.45, 3.01) | 1.2 | 73.71(71.84, 75.69) | 43.07(39.60, 46.42) | 43.48(40.61, 46.20) | 3.34(3.20, 6.35) | 2.04(1.79, 3.69) | 1.6 |

Table 3: Ablation study on the contribution of each module in FEMALA. We compare against a vanilla Transformer ('TF') and evaluate our key components: the Segment-aware Temporal Encoder (STE), the Structured Missingness Encoder (SME), naive concatenation ('Concat'), and our proposed Missing-pattern-guided Adaptive Fusion (MAF).

| | Model | IHM | | | | | READM | | | | |
|---|---|---|---|---|---|---|---|---|---|---|---|
| | | AUROC (↑) | AUPR (↑) | F1 (↑) | EO (↓) | EDDI (↓) | AUROC (↑) | AUPR (↑) | F1 (↑) | EO (↓) | EDDI (↓) |
| Pre-training | TF only | 79.09 | 37.45 | 40.56 | 10.22 | 3.47 | 71.48 | 38.03 | 41.84 | 9.59 | 4.70 |
| | SME only | 69.53 | 23.59 | 31.73 | 8.10 | 3.51 | 63.93 | 27.52 | 34.31 | 8.94 | 5.32 |
| | STE only | 83.34 | 48.14 | 47.48 | 8.21 | 3.38 | 75.87 | 46.96 | 45.92 | 9.08 | 5.53 |
| | TF+SME+MAF | 80.32 | 39.97 | 42.00 | 7.39 | 3.46 | 72.20 | 38.31 | 40.46 | 9.02 | 5.58 |
| | STE+SME+Concat. | 83.32 | 46.51 | 46.31 | 7.91 | 3.33 | 75.90 | 45.19 | 45.11 | 9.20 | 5.75 |
| | STE+SME+MAF | **84.59** | **48.22** | **47.88** | **6.78** | **3.28** | **76.49** | **47.00** | **46.34** | **8.85** | **4.60** |
| Fine-tuning | TF only | 78.78 | 37.26 | 41.61 | 7.68 | 2.74 | 72.20 | 38.73 | 42.99 | 5.94 | 3.03 |
| | SME only | 69.48 | 23.77 | 31.90 | 7.31 | 2.39 | 62.31 | 26.72 | 34.07 | 7.22 | 5.18 |
| | STE only | 83.24 | 46.14 | 48.08 | 7.02 | 2.15 | 74.08 | **45.71** | 44.03 | 6.69 | 3.69 |
| | TF+SME+MAF | 80.38 | 39.71 | 42.79 | 6.22 | 2.68 | 72.32 | 38.26 | 42.37 | 6.81 | 3.81 |
| | STE+SME+Concat. | 82.27 | 45.90 | 46.47 | 6.40 | 2.13 | 73.26 | 42.38 | 43.40 | 6.14 | 3.15 |
| | STE+SME+MAF | **83.49** | **46.66** | **48.18** | **5.37** | **1.95** | **74.60** | 45.42 | **44.55** | **4.92** | **2.80** |

some performance gains may appear modest, statistical significance tests in Appendix E.2 confirm that our improvements are statistically significant ($p < 0.05$) in most cases.

**Impact of Missingness Modeling on Subgroups.** To directly validate our hypothesis that structured missingness is linked to demographic bias, we analyze the performance gains from our Structured Missingness Encoder (SME) across different racial subgroups. As shown in Figure 4, the SME provides the most significant AUROC improvements for minority groups, particularly for HISPANIC patients (+3.49). This finding strongly suggests that our model successfully captures non-random, subgroup-specific missingness patterns—potentially reflecting disparities in care—which are ignored by other models. This analysis not only explains the source of our model's superior performance but also empirically establishes a crucial link between structured missingness and algorithmic fairness.

**Qualitative Analysis.** Figure 3 presents t-SNE visualizations of the learned representations. Compared to baselines that exhibit clear subgroup clustering (e.g., FairEHR-CLP, FLMD), FEMALA produces a more homogeneous latent space, especially after fine-tuning. This visually confirms the effectiveness of our adversarial tuning in removing sensitive information while maintaining a coherent representation space. Figure 5 further details the EO scores across all sensitive attributes. FEMALA consistently achieves lower (better) EO values, with the most pronounced challenges remaining in mitigating biases related to *Race* and *Age*, reflecting deep-rooted systemic disparities.

## 4.3 ABLATION STUDIES AND COMPARATIVE ANALYSIS

### 4.3.1 IMPACT OF ARCHITECTURAL COMPONENTS

We evaluate the contribution of each component in FEMALA in Table 3. The Segment-aware Temporal Encoder (STE) alone significantly outperforms a vanilla Transformer, confirming its effectiveness.

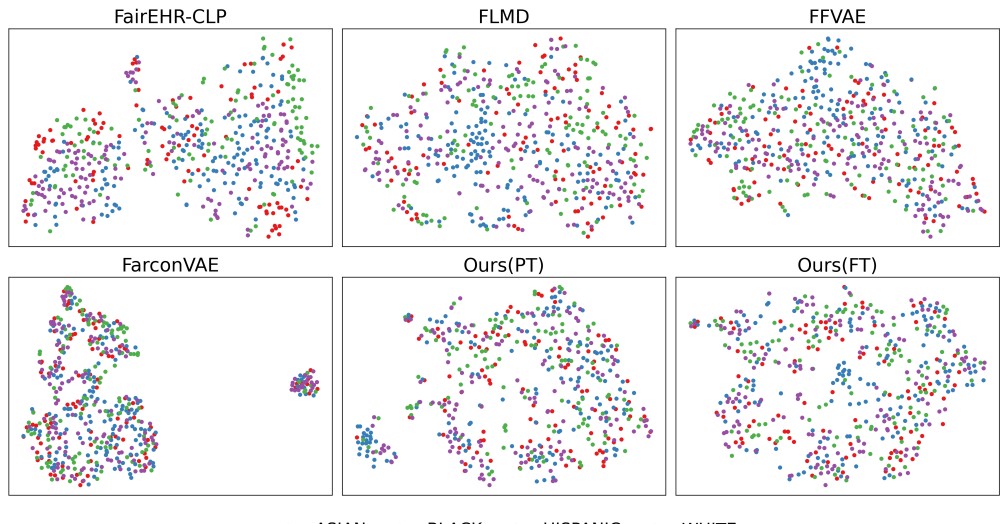

Figure 3: The t-SNE visualization of the learned representations on the MIMIC-III dataset.

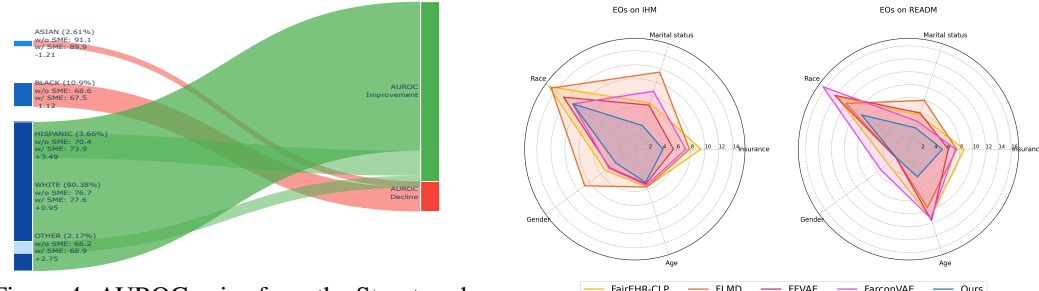

Figure 4: AUROC gains from the Structured Missingness Encoder (SME) across racial subgroups, highlighting the disproportionate benefit for minority groups.

Figure 5: The Equalized Odds of `FEMALA` on sensitive attributes on MIMIC-III dataset.

While the Structured Missingness Encoder (SME) offers modest gains independently, its true value is realized through principled integration. Notably, naive concatenation ('STE+SME+Concat') degrades performance, validating the necessity of our proposed Missingness-guided Adaptive Fusion (MAF) module ('STE+SME+MAF'), which yields the best overall performance. Finally, the results consistently show that adversarial low-rank fine-tuning effectively enhances fairness (lower EO/EDDI) with minimal impact on predictive accuracy, demonstrating its robustness.

### 4.3.2 JUSTIFICATION FOR DESIGN CHOICES AND STRATEGY COMPARISONS

To further validate our methodological choices, we compare our key strategies against strong alternatives.

**Missing Data Handling.** As shown in Table 4, we compare our explicit missingness modeling against standard imputation-based approaches. Our method significantly outperforms all alternatives, including powerful imputation models like SAITS and CSDI. This highlights that treating structured missingness as an informative signal is more effective than treating it as noise to be imputed or masked.

**Fairness-aware Training.** Table 5 confirms the superiority of our two-stage, low-rank adaptation strategy. One-stage adversarial training proves unstable, degrading both fairness and accuracy. While full-model fine-tuning restores performance, it fails to mitigate bias effectively. In contrast, our decoupled approach ('Adv. Low-rank FT') achieves the best trade-off, preserving task-specific knowledge while precisely mitigating bias.

Table 4: Comparison of missing data handling strategies on the MIMIC-III dataset.

| | Model | IHM | | | | | READM | | | | |
|---|---|---|---|---|---|---|---|---|---|---|---|
| | | AUROC (↑) | AUPR (↑) | F1 (↑) | EO (↓) | EDDI (↓) | AUROC (↑) | AUPR (↑) | F1 (↑) | EO (↓) | EDDI (↓) |
| Pre-training | Concat. | 80.40 | 37.59 | 40.37 | 9.59 | 4.00 | 71.26 | 36.98 | 40.85 | 9.82 | 5.16 |
| | Zero IMP. | 79.31 | 39.42 | 39.83 | 9.09 | 4.67 | 72.76 | 41.58 | 41.61 | 10.10 | 6.64 |
| | SAITS Du et al. (2023) | 82.69 | 45.41 | 43.35 | 7.34 | 4.09 | 74.83 | 44.34 | 43.27 | 9.57 | 5.02 |
| | CSDI Tashiro et al. (2021) | 82.23 | 43.99 | 41.91 | 7.77 | 3.92 | 74.37 | 43.27 | 41.40 | 9.97 | 5.52 |
| | Ours(PT) | **84.59** | **48.22** | **47.88** | **6.78** | **3.28** | **76.49** | **47.00** | **46.34** | **8.85** | **4.60** |
| Fine-tuning | Concat. | 80.01 | 38.44 | 41.83 | 7.60 | 2.32 | 71.49 | 37.66 | 41.58 | 7.55 | 4.15 |
| | Zero IMP. | 74.74 | 36.15 | 39.30 | 5.60 | 2.97 | 68.33 | 37.07 | 39.31 | 5.48 | 3.19 |
| | SAITS Du et al. (2023) | 81.71 | 43.63 | 43.61 | 5.97 | 3.37 | 72.90 | 42.42 | 41.84 | 5.61 | 5.02 |
| | CSDI Tashiro et al. (2021) | 81.09 | 41.86 | 42.22 | 5.71 | 3.54 | 72.41 | 41.25 | 40.02 | 5.76 | 5.52 |
| | Ours(FT) | **83.49** | **46.66** | **48.18** | **5.37** | **1.95** | **74.60** | **45.42** | **44.55** | **4.92** | **2.80** |

Table 5: Comparison of different fairness-aware training strategies on the MIMIC-III dataset.

| Model | IHM | | | | | READM | | | | |
|---|---|---|---|---|---|---|---|---|---|---|
| | AUROC (↑) | AUPR (↑) | F1 (↑) | EO (↓) | EDDI (↓) | AUROC (↑) | AUPR (↑) | F1 (↑) | EO (↓) | EDDI (↓) |
| PT | 84.59 | 48.22 | 47.88 | 6.78 | 3.28 | 76.49 | 47.00 | 46.34 | 8.85 | 4.60 |
| Adv. One-stage Training | 82.80 | 47.64 | 46.98 | 9.11 | 3.17 | 74.91 | 45.75 | 41.95 | 8.16 | 4.33 |
| Adv. Full Model FT | 84.39 | 47.65 | 47.29 | 8.95 | 3.20 | 74.76 | 43.43 | 42.98 | 5.56 | 3.63 |
| Adv. Low-rank FT | 83.49 | 46.66 | 48.18 | **5.37** | **1.95** | 74.60 | 45.42 | 44.55 | **4.92** | **2.80** |

**Segmentation Strategy.** To address a key concern from prior reviews, we empirically justify our use of fixed-length segmentation. Table 6 shows that our fixed-length approach outperforms both variable-length and event-based Transformer encoders. We attribute this to its better stability and effectiveness in capturing local temporal patterns in our datasets, confirming our design choice.

Table 6: Compared with variable-length segments and event-based Transformer encoders.

| | Model | IHM | | | | | READM | | | | |
|---|---|---|---|---|---|---|---|---|---|---|---|
| | | AUROC (↑) | AUPR (↑) | F1 (↑) | EO (↓) | EDDI (↓) | AUROC (↑) | AUPR (↑) | F1 (↑) | EO (↓) | EDDI (↓) |
| PT | Variable-length segments | 83.64 | 46.10 | 46.23 | 7.70 | 3.95 | 74.20 | 45.03 | 45.25 | 9.04 | 3.93 |
| | Event-based Transformer encoder | 83.60 | 46.40 | 46.20 | 9.16 | 3.71 | 68.81 | 40.91 | 40.24 | 9.39 | 6.25 |
| | Ours(PT) | 84.59 | 48.22 | 47.88 | 6.78 | 3.28 | 76.49 | 47.00 | 46.34 | 8.85 | 4.60 |
| FT | Variable-length segments | 81.92 | 45.32 | 45.39 | 6.29 | 2.74 | 71.50 | 43.63 | 43.70 | 4.99 | 3.57 |
| | Event-based Transformer encoder | 82.51 | 43.47 | 44.12 | 7.03 | 3.07 | 66.46 | 38.74 | 38.74 | 5.34 | 4.11 |
| | Ours (FT) | 83.49 | 46.66 | 48.18 | 5.37 | 1.95 | 74.60 | 45.42 | 44.55 | 4.92 | 2.80 |

## 5 CONCLUSION

We present FEMALA, a two-stage framework designed to mitigate demographic bias and effectively handle missing data in clinical outcome prediction. In the first stage, we combine a segment-aware time-series encoder with a structured missingness encoder to capture representations that reflect both the observed data and the underlying missingness patterns. In the second stage, we apply adversarial fine-tuning with low-rank adapters to encourage the model to retain task-relevant signals while minimizing reliance on sensitive attributes. We evaluate FEMALA on the MIMIC-III and MIMIC-IV datasets, where it achieves state-of-the-art performance in both predictive accuracy and fairness.

**Limitations and Future Work.** This work focuses on binary classification tasks using multivariate time-series data and categorical sensitive attributes. Future extensions may include adapting FEMALA to handle multi-modal clinical inputs—such as text and medical images—with more complex missingness patterns, as well as accommodating continuous or high-cardinality sensitive variables. Moreover, exploring alternative fine-tuning strategies, such as Direct Preference Optimization (DPO), may offer additional improvements in the fairness–accuracy trade-off.

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

APPENDIX OVERVIEW

This appendix provides supplementary material to support the main paper. We first clarify the role of Large Language Models (LLMs) and outline our reproducibility resources in Section A and Section B, respectively. We then provide additional details on the datasets, prediction tasks, and fairness metrics in Section C. Following that, Section D describes the implementation details, hyperparameter settings, and baseline methods. Finally, Section E presents additional experimental results, including further baseline comparisons, statistical significance tests, and computational cost analysis.

## A   THE USE OF LARGE LANGUAGE MODELS (LLMs)

Large Language Models (LLMs) were used in this work solely for language polishing and improving the clarity of writing. No LLMs contributed to research ideation, experimental design, analysis, or the development of core scientific content. All conceptual and technical contributions are original to the authors.

## B   REPRODUCIBILITY STATEMENT

To ensure the reproducibility of our work, we provide a comprehensive set of resources. Our full implementation is available as an anonymous code repository, linked in the abstract, which includes all necessary code and usage instructions to replicate our experiments.

## C   ADDITIONAL INFORMATION ON DATASETS AND TASKS

We extract five categories of continuous-valued clinical predictors from the MIMIC-III and MIMIC-IV datasets: vital signs, blood gases, renal function, metabolic panel, and hematology. We focus exclusively on the first 48 hours of patient data recorded after ICU admission, sampling observations at hourly intervals. Admissions with fewer than 48 hours of recorded data are excluded. Detailed predictor information is summarized in Table 7. We preprocess the raw data using the pipeline proposed by Harutyunyan et al. (2019). Notably, due to the extreme sparsity of arterial oxygen pressure data in MIMIC-IV, we include this predictor only for the MIMIC-III dataset. Consequently, the total number of predictors is 26 for MIMIC-III and 25 for MIMIC-IV.

Table 7: Summary of clinical predictors in longitudinal data for MIMIC-III/IV datasets, $^*$ indicates the predictor is only available in MIMIC-III.

| Category | Predictors |
|---|---|
| Vital Signs | Heart Rate, Systolic Blood Pressure, Diastolic Blood Pressure, Mean Blood Pressure, Respiratory Rate, Body Temperature, Oxygen Saturation |
| Blood Gases | Arterial Base Excess, Arterial Carbon Dioxide Pressure, Arterial Oxygen Pressure$^*$, Arterial pH |
| Renal Function | Blood Urea Nitrogen, Creatinine |
| Metabolic Panel | Ionized Calcium, Serum Chloride, Serum Glucose, Fingerstick Glucose, Anion Gap, Serum Bicarbonate, Magnesium, Serum Potassium, Serum Sodium |
| Hematology | Serum Hematocrit, Hemoglobin, Platelet Count, White Blood Cell Count |

We also extract five sensitive attributes from the MIMIC-III and MIMIC-IV datasets: insurance type, marital status, race, gender, and age. Each attribute contains several subgroups, whose compositions vary between the two datasets. For instance, the insurance attribute comprises five subgroups (*Medicare, Medicaid, Government, Self Pay, and Private*) in MIMIC-III, whereas it includes only three subgroups (*Medicare, Medicaid, and Other*) in MIMIC-IV due to differences in recording standards. The distributions of sensitive attributes across MIMIC-III and MIMIC-IV are presented in Figure 6. Notably, the subgroup distributions in MIMIC-III exhibit greater imbalance compared to those in MIMIC-IV, reflecting the more diverse subgroup categorization in certain sensitive attributes.

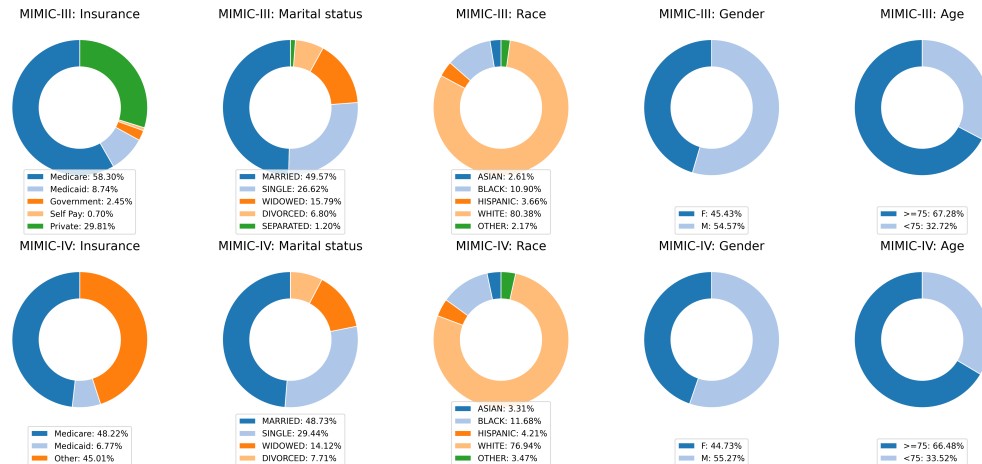

Figure 6: Distribution of sensitive attributes in MIMIC-III and MIMIC-IV datasets.

## C.1 TASKS

**In-Hospital Mortality (IHM) Prediction.** The In-Hospital Mortality (IHM) task involves predicting whether a patient will pass away during their hospital stay. As summarized in Table 1, the MIMIC-III dataset contains 1,484 positive samples in the training set with a missing data rate of 72.81%. Similarly, the MIMIC-IV dataset includes 1,702 positive samples in the training set, with a slightly lower missing data rate of 71.16%. **Readmission (READM) Prediction.** The Readmission (READM) task aims to forecast whether a patient will be readmitted to the hospital within 30 days after discharge. In this task, both patients who are readmitted and those who pass away during hospitalization are treated as positive cases. As shown in Table 1, the MIMIC-III dataset contains 2,268 positive samples in the training set with a missing data rate of 72.81%. The MIMIC-IV dataset has 2,648 positive samples in the training set, and a missing data rate of 71.16%.

## C.2 FAIRNESS METRICS

**Equalized Odds (EO)** is a widely adopted group-fairness metric that measures disparities in prediction errors across different subgroups. Specifically, EO quantifies the discrepancy in True Positive Rates (TPR) and False Positive Rates (FPR) between privileged and unprivileged groups. Lower EO values signify more equitable model predictions.

Given multiple sensitive attributes, each containing multiple subgroups, we calculate EO by averaging pairwise disparities across all subgroup pairs within each attribute. Formally, the EO for each sensitive attribute is computed as:

$$\text{EO}_{\text{TPR}} = \frac{1}{\binom{|A|}{2}} \sum_{a_i} \sum_{a_j > a_i} \big| \text{TPR}_{a_i} - \text{TPR}_{a_j} \big|, \tag{14}$$

$$\text{EO}_{\text{FPR}} = \frac{1}{\binom{|A|}{2}} \sum_{a_i} \sum_{a_j > a_i} \big| \text{FPR}_{a_i} - \text{FPR}_{a_j} \big|. \tag{15}$$

Here, $A$ denotes the set of subgroups within a sensitive attribute (e.g., *Insurance* = {Medicaid, Medicare, ... }), and $\binom{|A|}{2}$ represents the total number of subgroup pairs. For each subgroup $a \in A$, we define:

$$\text{TPR}_a = \frac{\text{TP}_a}{\text{TP}_a + \text{FN}_a}, \qquad \text{FPR}_a = \frac{\text{FP}_a}{\text{FP}_a + \text{TN}_a},$$

where $\text{TP}_a$, $\text{FP}_a$, $\text{FN}_a$, and $\text{TN}_a$ represent the subgroup-specific confusion matrix components.

The EO score for each sensitive attribute is then obtained by averaging the disparities in TPR and FPR:

$$\text{EO} = \frac{1}{2}\left(\text{EO}_{\text{TPR}} + \text{EO}_{\text{FPR}}\right).$$

The overall EO is finally computed by averaging across all sensitive attributes.

**Error Distribution Disparity Index (EDDI)** complements EO by quantifying the consistency of prediction errors across demographic groups. Following Wang et al. (2024c), EDDI measures the extent to which the subgroup-specific error rates deviate from the overall error rate. Formally, given a set of subgroups $A$ within a sensitive attribute and a subgroup $a \in A$:

$$\text{EDDI} = \frac{1}{|A|} \sum_{a \in A} \frac{|\text{ER}_a - \text{OER}|}{\max(\text{OER}, 1 - \text{OER})},$$

where subgroup error rate ($\text{ER}_a$) and overall error rate (OER) are defined as:

$$\text{ER}_a = \frac{1}{N_a} \sum_{i \in a} \mathbb{1}(y_i \neq \hat{y}_i), \qquad \text{OER} = \frac{1}{N} \sum_{i=1}^{N} \mathbb{1}(y_i \neq \hat{y}_i).$$

Here, $y_i$ and $\hat{y}_i$ are the true and predicted labels, respectively, $N_a$ is the number of samples in subgroup $a$, and $N$ is the total dataset size. EDDI normalizes each subgroup's error-rate deviation by the maximum possible error deviation ($\max(\text{OER}, 1 - \text{OER})$), providing a standardized measure of fairness. Lower EDDI values indicate greater fairness, reflecting more uniform prediction accuracy across demographic subgroups.

# D  MORE ON BASELINE METHODS AND IMPLEMENTATION DETAILS

## D.1  IMPLEMENTATION DETAILS AND HYPERPARAMETERS

**Implementation Details.** FEMALA is implemented in Python 3.11 using PyTorch 2.0. All experiments are conducted on a single NVIDIA RTX-4090 GPU with a batch size of 128. Models are trained for a maximum of 100 epochs using the Adam optimizer, with early stopping triggered if validation AUROC does not improve for 10 consecutive epochs. The best-performing model on the validation set is selected for final evaluation on the test set. For calculating F1 score and fairness metrics, we use the threshold that yields the best F1 score on the validation set. For adversarial fine-tuning, we initiate the process after pre-training the MINE module for 30 epochs. For all baselines, we concatenate the time-series data with its corresponding missingness mask as the model input.

**Hyperparameter Tuning.** We tune key hyperparameters via grid search on the validation set. The search spaces are as follows:

- Dropout ratio: $\{0, 0.1, 0.2, 0.3\}$
- Learning rate: $\{1 \times 10^{-4}, 5 \times 10^{-5}, 1 \times 10^{-5}\}$

All reported results use the optimal hyperparameter settings identified through this process.

**Sensitivity Analysis of Segment Length ($L$) and LoRA Rank ($r$).** We investigate the sensitivity of FEMALA to two key hyperparameters: segment length $L$ and LoRA rank $r$. As shown in Figure 7, shorter segments ($L$) slightly improve performance on IHM, while longer segments benefit READM, suggesting that optimal temporal resolution is task-specific. Figure 8 indicates that increasing the LoRA rank $r$ enhances predictive performance but may also amplify bias. Moderate values (e.g., $r = 8$) offer the best trade-off, achieving strong accuracy while maintaining fairness.

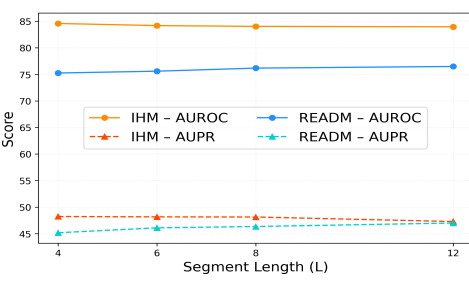
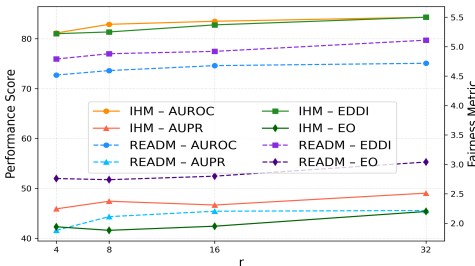

Figure 7: Effect of segment length $L$.                    Figure 8: Effect of rank $r$.

**Sensitivity Analysis of the Mutual Information Coefficient ($\lambda_{\mathbf{MI}}$).** We investigate the impact of the mutual information regularization coefficient, $\lambda_{\mathrm{MI}}$, on the trade-off between predictive performance and fairness. As shown in Table 8, we vary $\lambda_{\mathrm{MI}}$ from 0.1 to 1.0 and observe its effect on the READM task. The results reveal a clear trend: as $\lambda_{\mathrm{MI}}$ increases, the model places greater emphasis on minimizing the mutual information between the specific and shared representations. This leads to improved fairness, as evidenced by the decreasing Equalized Odds (EO) and Error Distribution Disparity Index (EDDI). However, this fairness enhancement comes at the cost of a slight reduction in predictive performance, reflected in the lower AUROC, AUPR, and F1 scores. Based on these findings, we selected $\lambda_{\mathrm{MI}} = 0.5$ for our main experiments, as it strikes an effective balance, achieving substantial fairness gains without excessively compromising predictive accuracy.

Table 8: Sensitivity analysis of the mutual information coefficient ($\lambda_{\mathrm{MI}}$).

| $\lambda_{\mathrm{MI}}$ | MIMIC-III READM | | | | |
|---|---|---|---|---|---|
| | AUROC (↑) | AUPR (↑) | F1 (↑) | EO (↓) | EDDI (↓) |
| PT | 76.49 | 47.00 | 46.34 | 8.85 | 4.60 |
| 0.1 | 76.39 | 46.88 | 46.58 | 7.95 | 4.39 |
| 0.2 | 75.58 | 46.14 | 45.75 | 6.98 | 4.17 |
| 0.5 | 74.60 | 45.42 | 44.55 | 4.92 | 2.80 |
| 0.7 | 74.44 | 44.73 | 44.21 | 4.90 | 2.81 |
| 1.0 | 74.02 | 43.92 | 43.81 | 4.87 | 2.77 |

## D.2 DETAILED DESCRIPTIONS OF BASELINE METHODS

**Baselines without fairness strategies:**

- **CNN** (LeCun et al., 1998): Convolutional Neural Networks utilize convolutional layers to automatically extract hierarchical representations, enabling the learning of complex decision boundaries for predictive tasks.

- **RNN** (Elman, 1990): Recurrent Neural Networks process sequential data by recursively passing hidden states through time steps, making them effective for modeling temporal dependencies.

- **LSTM** (Graves and Graves, 2012): Long Short-Term Memory networks are specialized recurrent architectures designed to effectively capture long-term dependencies and mitigate the vanishing gradient problem inherent in standard RNNs.

- **Transformer** (Vaswani et al., 2017): Transformer architectures leverage self-attention mechanisms, allowing models to efficiently capture global dependencies without recurrent connections, thus demonstrating excellent generalization across multiple domains.

**Baselines with fairness strategies:**

- **FFVAE** (Creager et al., 2019): It employs adversarial decorrelation within a variational autoencoder framework to disentangle sensitive attributes from latent representations, ensuring fairness in downstream predictions.

- **FarconVAE** (Oh et al., 2022): It combines variational autoencoder techniques with contrastive learning objectives to achieve fair representation learning through disentanglement.

- **FairEHR-CLP** (Wang et al., 2024c): This model integrates generative adversarial networks for synthesizing counterfactual patient data, followed by contrastive learning to explicitly reduce prediction biases across demographic groups.

- **FLMD** (Liu et al., 2023): It employs deconfounder theory to infer and incorporate latent confounders, improving fairness by addressing unobserved biases within the dataset.

**Imputation Baselines:**

- **SAITS** (Du et al., 2023): Self-Attention-based imputation for Time Series utilizes transformer-based architectures with self-attention mechanisms to effectively impute missing values in multivariate time series.

- **CSDI** (Tashiro et al., 2021): Diffusion-based imputation adopts diffusion probabilistic models to provide robust and probabilistically sound imputation for multivariate time-series data.

# E MORE ON EXPERIMENTAL RESULTS

## E.1 EXPERIMENTS ON ADDITIONALLY BASELINES

To further validate the effectiveness of FEMALA, we conduct additional experiments comparing it with more baseline methods on the MIMIC-III dataset, including Adversarial Training (Yang et al., 2023), Adapting Fairness (Feng et al., 2023), and FairLoRA (Sukumaran et al., 2024). The results, presented in Table 9, demonstrate that FEMALA consistently outperforms these baselines in both predictive performance and fairness metrics. This further substantiates the robustness and efficacy of our proposed approach in achieving fair and accurate predictions in clinical settings.

Table 9: Performance and fairness evaluation across two tasks on MIMIC-III.

| Model | In-Hospital Mortality | | | | | Readmission | | | | |
|---|---|---|---|---|---|---|---|---|---|---|
| | AUROC ($\uparrow$) | AUPR ($\uparrow$) | F1 ($\uparrow$) | EO ($\downarrow$) | EDDI ($\downarrow$) | AUROC ($\uparrow$) | AUPR ($\uparrow$) | F1 ($\uparrow$) | EO ($\downarrow$) | EDDI ($\downarrow$) |
| Adversarial Training | 82.25 | 42.55 | 43.45 | 7.66 | 4.03 | 72.41 | 40.18 | 41.43 | 7.43 | 5.88 |
| Adapting Fairness | 80.46 | 42.14 | 43.22 | 8.42 | 5.08 | 71.96 | 41.21 | 41.31 | 8.14 | 6.02 |
| FairLoRA | 78.95 | 37.79 | 41.68 | 7.13 | 2.88 | 72.53 | 38.93 | 42.94 | 5.41 | 3.51 |
| Ours(PT) | 84.59 | 48.22 | 47.88 | 6.78 | 3.28 | 76.49 | 47.00 | 46.34 | 8.85 | 4.60 |
| Ours (FT) | 83.49 | 46.66 | 48.18 | 5.37 | 1.95 | 74.60 | 45.42 | 44.55 | 4.92 | 2.80 |

## E.2 STATISTICAL SIGNIFICANCE ANALYSIS

To rigorously evaluate the statistical significance of our results, we conduct two-sample bootstrapped $t$-tests with a significance level of 0.05. This analysis provides a detailed comparison of FEMALA against various benchmarks and internal configurations, with the results summarized in Tables 10 to 13. Metrics highlighted in red indicate instances where FEMALA is outperformed.

**Comparison with SOTA Baselines.** As shown in Table 10, we compare FEMALA against both fair and non-fair state-of-the-art baselines. The results demonstrate that FEMALA achieves statistically significant improvements in most performance and fairness metrics. In the few cases where a baseline shows slightly better predictive performance (e.g., on MIMIC-IV), FEMALA consistently delivers statistically superior fairness, underscoring its effectiveness in balancing accuracy and equity.

**Ablation Study on Model Components.** Table 11 presents an ablation study that assesses the contribution of each module within FEMALA. The p-values confirm that each component provides a statistically significant benefit to the model's overall performance and fairness, validating our design choices.

**Comparison of Missing Data Handling Strategies.** In Table 12, we compare our proposed strategy for handling missing data against other state-of-the-art imputation methods. The results show that our

approach leads to statistically significant gains in both predictive accuracy and fairness, highlighting its superiority.

**Comparison of Fairness-Aware Training Strategies.** Finally, Table 13 compares our adversarial fine-tuning strategy with alternative fairness-aware training methods. The analysis confirms that our approach is significantly more effective at enhancing fairness, often while maintaining or improving predictive performance.

Collectively, these statistical tests validate that FEMALA not only surpasses existing methods but also benefits significantly from its unique architectural components and training strategies.

Table 10: P-values of two-sample bootstrapped $t$-test of FEMALA compared to SOTA non-fair / fair SOTAs. We observe that most of the tests are significant under 0.05 significance level.

| Model | IHM | | | | | READM | | | | |
|---|---|---|---|---|---|---|---|---|---|---|
| | AUROC (↑) | AUPR (↑) | F1 (↑) | EO (↓) | EDDI (↓) | AUROC (↑) | AUPR (↑) | F1 (↑) | EO (↓) | EDDI (↓) |
| MIMIC-III v.s. non-fair | 3.16E-125 | 1.50E-66 | 2.05E-105 | / | / | 1.40E-106 | 1.69E-98 | 2.04E-107 | / | / |
| MIMIC-III v.s. fair | 4.20E-41 | 2.79E-121 | 4.32E-140 | 9.82E-37 | 8.70E-16 | 6.08E-17 | 2.30E-96 | 1.81E-64 | 2.92E-99 | 1.31E-29 |
| MIMIC-IV v.s. non-fair | 3.11E-09 | 0.025 | 4.44E-16 | / | / | 1.01E-26 | 1.06E-20 | 1.29E-12 | / | / |
| MIMIC-IV v.s. fair | 0.0081 | 9.57E-14 | 2.62E-42 | 5.77E-43 | 0.26 | 4.33E-09 | 0.00022 | 8.22E-05 | 2.15E-20 | 5.68E-16 |

Table 11: P-values of two-sample bootstrapped $t$-test for the ablation study on model components, comparing the full FEMALA model against its variants.

| | Model | IHM | | | | | READM | | | | |
|---|---|---|---|---|---|---|---|---|---|---|---|
| | | AUROC (↑) | AUPR (↑) | F1 (↑) | EO (↓) | EDDI (↓) | AUROC (↑) | AUPR (↑) | F1 (↑) | EO (↓) | EDDI (↓) |
| PT | FEMALA w/ STE vs FEMALA w/ TF | 5.51E-264 | 1.25E-190 | 3.07E-198 | / | / | 3.92E-235 | 4.92E-258 | 1.65E-260 | / | / |
| | FEMALA vs. STE | 1.22E-42 | 3.73E-01 | 1.81E-02 | / | / | 4.41E-10 | 3.90E-01 | 3.02E-03 | / | / |
| | FEMALA vs. STE+SME+Concat. | 2.47E-43 | 3.88E-14 | 3.60E-19 | / | / | 6.16E-09 | 2.68E-20 | 8.20E-21 | / | / |
| FT | FEMALA w/ STE vs FEMALA w/ TF | 7.55E-175 | 2.42E-152 | 5.73E-183 | 2.82E-49 | 1.50E-74 | 6.23E-91 | 4.26E-203 | 3.04E-48 | 3.58E-54 | 1.23E-50 |
| | FEMALA vs. STE | 9.17E-02 | 2.29E-02 | 3.28E-01 | 1.50E-91 | 1.06E-04 | 7.02E-07 | 1.19E-01 | 5.07E-04 | 3.10E-56 | 8.76E-39 |
| | FEMALA vs. STE+SME+Concat. | 4.82E-38 | 7.58E-04 | 1.34E-25 | 2.19E-18 | 7.88E-04 | 6.70E-36 | 3.48E-53 | 2.76E-17 | 5.05E-24 | 4.64E-06 |

Table 12: P-values of two-sample bootstrapped $t$-test compare the SOTA missing data handling strategies with FEMALA.

| Model | IHM | | | | | READM | | | | |
|---|---|---|---|---|---|---|---|---|---|---|
| | AUROC (↑) | AUPR (↑) | F1 (↑) | EO (↓) | EDDI (↓) | AUROC (↑) | AUPR (↑) | F1 (↑) | EO (↓) | EDDI (↓) |
| PT | 3.09E-92 | 2.22E-35 | 3.66E-135 | / | / | 1.70E-60 | 4.41E-41 | 1.22E-102 | / | / |
| FT | 7.19E-79 | 3.05E-40 | 1.08E-135 | 4.80E-02 | 3.82E-11 | 4.36E-55 | 8.26E-52 | 9.47E-81 | 1.52E-05 | 1.58E-05 |

Table 13: P-values of two-sample bootstrapped $t$-test compare different fairness-aware training strategies with FEMALA.

| Model | IHM | | | | | READM | | | | |
|---|---|---|---|---|---|---|---|---|---|---|
| | AUROC (↑) | AUPR (↑) | F1 (↑) | EO (↓) | EDDI (↓) | AUROC (↑) | AUPR (↑) | F1 (↑) | EO (↓) | EDDI (↓) |
| Adv. One-stage Training | 8.69E-14 | 1.82E-05 | 9.27E-13 | 1.17E-150 | 8.07E-86 | 3.04E-03 | 8.83E-02 | 2.88E-78 | 1.30E-131 | 6.19E-64 |
| Adv. Full Model FT | 4.54E-24 | 1.10E-05 | 2.96E-07 | 5.65E-143 | 1.31E-116 | 1.03E-01 | 1.76E-25 | 1.77E-30 | 1.48E-07 | 1.70E-26 |

## E.3 COMPUTATIONAL COST ANALYSIS

We provide a detailed analysis of the computational costs for all models in Table 14, including parameter counts and training times. The results demonstrate that FEMALA achieves superior performance and fairness while maintaining computational costs comparable to the baselines, highlighting its practicality and scalability.

Table 14: Computational cost compare.

| Model | Trainable Params | Total Params | Training time per epoch | Converge Epoch |
|---|---|---|---|---|
| Transformer | 13.77 MB | 13.77 MB | 9.22 s | 15 |
| LSTM | 2.88 MB | 2.88 MB | 7.16 s | 19 |
| RNN | 0.72 MB | 0.72 MB | 7.24 s | 25 |
| CNN | 0.51 MB | 0.51 MB | 6.50 s | 16 |
| FairEHR-CLP | 14.01 MB | 14.01 MB | 6.14 s | 22 |
| FLMD | 1.54 MB / 18.56 MB | 20.10 MB | 21.16 s / 15.51 s | 163/22 |
| FFVAE | 0.75 MB / 0.13 MB | 0.88 MB | 9.40 s / 7.15 s | 100/10 |
| FarconVAE | 1.89 MB / 0.0005 MB | 1.89 MB | 13.46 s / 7.18 s | 300/14 |
| FEMALA(PT) | 15.54 MB | 15.54 MB | 13.72 s | 30 |
| FEMALA(FT) | 2.61 MB | 18.14 MB | 11.16 s | 35 |

