# OpenReview forum: "Fairness-Aware EHR Analysis via Structured Missing Pattern Modeling and Adversarial Low-Rank Adaptation"
_ICLR.cc/2026/Conference — ICLR 2026 Conference Withdrawn Submission_

### Official Review · Reviewer_ueUL · 2025-10-20

**Soundness:** 1
**Presentation:** 2
**Contribution:** 1
**Rating:** 0
**Confidence:** 4

**Summary:**

This paper aims to address the issues of fairness and data missingness in EHR data simultaneously. Overall, the research topic is not novel, and its significance is limited for a top machine learning conference. Both fairness and missingness have been extensively studied in existing works on AI in healthcare, while this paper mainly attempts to combine these two aspects. The novelty of the proposed “ensemble” method is limited. In addition, the baselines used for comparison are not adequate, and the evaluation focuses only on naive empirical studies, without any simulated settings that include different missingness mechanisms or synthetic data where the ground truth is known.

I suggest that the authors revise the article to address the following weaknesses and consider submitting it to a clinical informatics journal as a general benchmark or empirical study.

Please find my detailed comments below.

**Strengths:**

The overall writing is clear and easy to follow, and the figures are well presented.

**Weaknesses:**

1.	Lack of novelty and significance.

As mentioned in the overall summary, the work lacks novelty. I suggest that the authors reconsider the target audience of this study and revise the manuscript for submission to a clinical informatics journal, where it may be of greater interest.

2.	Experimental design.

The experimental design requires substantial improvement to meet the standards of a rigorous study.

(1) Baseline models

The baseline models are not sufficient. More deep learning–based models should be included, following prior works. For example:

(a) Liu, Mingxuan, et al. “Handling missing values in healthcare data: A systematic review of deep learning-based imputation techniques.” Artificial Intelligence in Medicine 142 (2023): 102587.

(b) Penny, Kay I., and Ian Atkinson. “Approaches for dealing with missing data in health care studies.” Journal of Clinical Nursing 21(19–20) (2012): 2722–2729.

State-of-the-art fairness methods for structured EHR data should also be considered:

(a) Berk, Richard, et al. "A convex framework for fair regression." arXiv preprint arXiv:1706.02409 (2017).

(b) Williamson, Robert, and Aditya Menon. "Fairness risk measures." International conference on machine learning. PMLR, 2019.



In addition, naive MVI (missing value imputation) strategies such as mean/median imputation and MICE should be compared. The proposed method must be clearly justified as offering unique advantages over these naive baselines. It is possible that standard MVI strategies could yield comparable or even better results for downstream tasks; if so, the proposed method would appear computationally expensive and overly black-box for clinical decision-making requirements.

(2) Simulations

Simulation studies are needed. The authors should first generate three different missingness mechanisms using real data, and ideally include experiments on purely simulated data as well.

(3) Reporting uncertainty

The paper should report standard errors or confidence intervals, since metrics such as EOR and DPR can fluctuate significantly when data are imbalanced.

**Questions:**

Could the authors provide more details to justify the unique challenges faced by temporal EHR data? This aspect is ambiguous in the current writing. The evaluation currently focuses only on prediction performance, where the outcome is treated as binary. For instance, if the goal is survival analysis, the current binary outcome prediction is insufficient. More experimental designs are needed to demonstrate that the proposed solution has unique advantages for time-series data. The current evaluation appears somewhat irrelevant to time-series data and may be misleading.

---

### Official Review · Reviewer_8e9B · 2025-10-31

**Soundness:** 3
**Presentation:** 3
**Contribution:** 2
**Rating:** 4
**Confidence:** 3

**Summary:**

The work proposes FEMALA (Fairness-Aware EHR Analysis via Structured Missing Pattern Modeling and Adversarial Low-Rank Adaptation), a two-stage framework for fair and accurate clinical outcome prediction from Electronic Health Records (EHRs). Recognizing that structured missingness in EHRs, non-random patterns of missing data correlated with demographic attributes, exacerbates algorithmic bias, FEMALA first jointly models observed clinical time-series data and structured missingness using a dual-encoder architecture with a novel adaptive fusion module. This enhances representation robustness and implicitly improves fairness. In the second stage, it applies adversarial fine-tuning via low-rank adaptation (LoRA) to minimize mutual information between task representations and sensitive attributes, effectively decoupling predictions from demographic bias while preserving accuracy. Evaluated on MIMIC-III and MIMIC-IV for in-hospital mortality and readmission prediction, FEMALA achieves state-of-the-art results in both AUROC (+2.3%) and fairness metrics like Equalized Odds (−2.9%), outperforming existing fairness-aware and imputation-based baselines.

**Strengths:**

1. A key strength of FEMALA is its recognition that missing data in Electronic Health Records (EHRs) is not random but systematically linked to demographic disparities, a phenomenon known as structured missingness.

2. FEMALA introduces a principled two-stage pipeline that separates representation learning from fairness enforcement, addressing a major limitation of existing methods that jointly optimize conflicting objectives.

**Weaknesses:**

1. The work restricts itself to binary classification tasks and categorical sensitive attributes, which narrows its applicability in real-world clinical settings where outcomes can be multi-class (e.g., disease severity levels) or continuous (e.g., length of stay), and where sensitive attributes may include continuous or high-cardinality variables such as socioeconomic status or zip code. This constraint limits the generalizability of FEMALA to more complex prediction scenarios commonly encountered in healthcare analytics and may require non-trivial architectural or methodological extensions to accommodate such cases.

2. Another weakness is the exclusive focus on multivariate time-series EHR data, excluding other rich modalities like clinical notes, imaging, or genomic data. Modern clinical decision support systems increasingly rely on multimodal inputs, and missingness patterns in these additional modalities could further compound fairness issues. By not addressing multimodal missingness or cross-modal bias propagation, the framework may overlook critical sources of disparity that arise when integrating heterogeneous data types, thus limiting its relevance in the context of contemporary multimodal clinical AI systems.

3. The evaluation, while thorough on MIMIC-III and MIMIC-IV, is confined to ICU populations from a single institution (Beth Israel Deaconess Medical Center), raising concerns about external validity. Algorithmic fairness is highly context-dependent, and biases manifest differently across healthcare systems, geographic regions, and patient demographics. Without validation on more diverse datasets—such as those from community hospitals, international cohorts, or outpatient settings—it remains uncertain whether FEMALA’s gains in fairness and accuracy would generalize beyond the specific structural and demographic characteristics of the MIMIC databases.

4. The fairness intervention in FEMALA assumes access to ground-truth sensitive attributes during fine-tuning, which may not always be available or legally permissible in practice due to privacy regulations or institutional policies. While the paper acknowledges this implicitly by using attributes like race and insurance status, it does not explore or propose alternatives for settings where such attributes are missing or must be inferred, nor does it address the ethical implications of requiring sensitive data to enforce fairness—a paradox that has been noted in prior fairness literature.

**Questions:**

Please refer to the weaknesses

---

### Official Review · Reviewer_Wp15 · 2025-10-31

**Soundness:** 3
**Presentation:** 3
**Contribution:** 2
**Rating:** 4
**Confidence:** 3

**Summary:**

This paper proposes FEMALA, a two-stage framework for fair EHR analysis that tackles bias from structured missingness. It uses a dual-encoder to model both temporal data and missing patterns, then applies adversarial low-rank adaptation (LoRA) to fine-tune for fairness, achieving state-of-the-art accuracy and fairness.

**Strengths:**

* **Missingness as Signal:** Innovatively treats missingness patterns as an informative signal, not just noise to be imputed.
* **Stable Two-Stage Design:** The "learn first, correct later" approach using adversarial LoRA achieves a superior accuracy-fairness trade-off.
* **Extensive Empirical Eval:** Performs a fairly extensive eval across baselines and datasets.

**Weaknesses:**

* **SOTA Claim** The method claims SOTA performance in a few places, yet the only recent baseline is one from 2024, which may not be the best comparison either (due to the focus of the other study on multimodal data).
* **Fixed Segmentation:** Relies on fixed-length segmentation, which may be less effective than event-based or adaptive methods.
* **Simple Missingness Encoding:** The method for encoding global missingness patterns (time/feature averages) is relatively simple.
* **Remaining Biases:** The model still struggles to fully mitigate deep-rooted biases related to sensitive attributes.

**Questions:**

- Do the authors claim that their results are better than FLMD and FFVAE in Fig. 3?

---

### Official Review · Reviewer_fUSL · 2025-11-01

**Soundness:** 3
**Presentation:** 3
**Contribution:** 2
**Rating:** 4
**Confidence:** 3

**Summary:**

The manuscript presents a two-stage framework for fairness aware prediction on EHR time series - FEMALA (Fairness-Aware EHR Analysis via Missing Pattern Modeling and Adversarial Low-Rank Adaptation). In Stage 1, the model jointly encodes EHR temporal dynamics with a Segment Aware Temporal Encoder and structured missingness with a Structured Missing Pattern Encoder. These are then fused. In Stage 2, the model learns an autoencoder based sensitive attribute embedding (gender, race, insurance, age, marital status), estimates mutual information between this embedding and the task representation using MINE. It then performs adversarial finetuning over LoRA adapters to minimize that MI while keeping task performance. In hospital mortality (IHM) and 30 days readmission (READM) are the two tasks used in the experiments on MIMIC-III / IV.

**Strengths:**

1. Clearly defined two stage system: Stage 1 creates a representation by jointly encoding EHR segments & structured masks. Stage 2 performs fairness specific adaptation. It connects two ideas of modeling of structured missingness and fairness aware adaptation.
2. Dual encoder with Structured Missingness Encoder and Missing Pattern Guided Adaptive Fusion module provides a principled way to turn structured missingness patterns into useful signal. The low rank adversarial finetuning that minimizes mutual information with sensitive attributes is a parameter efficient strategy that preserves accuracy while improving fairness.
3. Results on MIMIC III/IV dataset and two clinical tasks (IHM, readmission) show overall gains in predictive performance and fairness. Includes ablations.

**Weaknesses:**

1. Combining missingness modeling with fairness is valuable - but these components draw on known ideas (Transformers for time series, cross attention fusion, MINE based adversarial training).
2. Empirical scope is narrow to make a generic conclusion - results are on two related MIMIC datasets and two binary tasks from the same health system.
3. Proposed MAF uses a learned gating and attention over mask informed weights. Exact dimensions and normalization of α is unclear.
4. Existing method assumes access to multiple categorical sensitive attributes (race, gender, insurance, age group, marital status) and encodes them. Although mentioned as future work but evaluating on using high cardinality / continuous attributes is important. From the real-world practicality standpoint, these sensitive attributes may not be made available - manuscript paper does not discuss settings where sensitive attributes are partially or not available.
5. Fairness results are presented for EO and EDDI (group metrics). No per patient or path level analysis is made available.
6. Limited comparisons to missingness as a signal baseline (dual stream or mask encoder approach)

**Questions:**

1. Could you please report train on MIMIC III => test on MIMIC IV (and vice versa) to evaluate whether fairness gains remain under dataset shift?
2. Since one stage adversarial training is unstable, could you please share training curves for Stage 2 and report variance over to show the fairness gains are robust?
3. Could you please quantify the dependence between SME features and sensitive attributes?

---

### Note · Authors · 2025-11-12

I have read and agree with the venue's withdrawal policy on behalf of myself and my co-authors.